# Detection of SARS-CoV-2 in subcutaneous fat but not visceral fat, and the disruption of fat lymphocyte homeostasis in both fat tissues in the macaque

Anaëlle Olivo [1], Romain Marlin [1], Thierry Lazure[2], Pauline Maisonnasse [1], Laetitia Bossevot[1], Christelliah Mouanga[1], Julien Lemaitre[1], Guillaume Pourcher[3,4], Stéphane Benoist[5], Roger Le Grand[1], Olivier Lambotte[1,6,7], Nathalie Dereuddre-Bosquet [1,7] & Christine Bourgeois [1,7 ✉]

The well documented association between obesity and the severity of SARS-CoV-2 infection raises the question of whether adipose tissue (AT) is impacted during this infection. Using a model of SARS-CoV-2 infection in cynomolgus macaques, we detected the virus within subcutaneous AT (SCAT) but not in visceral AT (VAT) or epicardial AT on day 7 post-infection. We sought to determine the mechanisms responsible for this selective detection and observed higher levels of angiotensin-converting-enzyme-2 mRNA expression in SCAT than in VAT. Lastly, we evaluated the immunological consequences of SARS-CoV-2 infection on AT: both SCAT and VAT T cells showed a drastic reduction in CD69 expression, a standard marker of resident memory T cell in tissue, that is also involved in the migratory and metabolic properties of T cells. Our results demonstrate that in a model of mild infection, SCAT is selectively infected by SARS-CoV-2 although changes in the immune properties of AT are observed in both SCAT and VAT.

[1] Université Paris-Saclay, CEA, INSERM UMR-1184, Center for Immunology of Viral, Auto-immune, Hematological and Bacterial Diseases (IMVA-HB/IDMIT), 94276 Le Kremlin-Bicêtre, France. [2] Assistance Publique-Hôpitaux de Paris, Hôpital Bicêtre, Service d'anatomo-pathologie, 94275 Le Kremlin Bicêtre, France. [3] Obesity Center, Department of Digestive, Oncologic and Metabolic Surgery, Institut Mutualiste Montsouris, 75014 Paris, France. [4] Centre de Recherche en Epidémiologie et Santé des Population (CESP), Inserm, Université Paris-Saclay, 94800 Villejuif, France. [5] Assistance Publique-Hôpitaux de Paris, GHU Paris-Saclay, Hôpital Bicêtre, Service de Chirurgie Digestive et Oncologique, 94275 Le Kremlin Bicêtre, France. [6] Université Paris-Saclay, Assistance Publique-Hôpitaux de Paris, GHU Paris-Saclay, Hôpital Bicêtre, Service de Médecine Interne et Immunologie Clinique, 94275 Le Kremlin Bicêtre, France. [7] These authors contributed equally: Olivier Lambotte, Nathalie Dereuddre-Bosquet, Christine Bourgeois. ✉email: christine.bourgeois@universite-paris-saclay.fr

The association between obesity and the severity of coronavirus disease 2019 (COVID-19, caused by severe acute respiratory syndrome coronavirus 2 (SARS-CoV-2)) is well documented;[1–7] along with age, male sex, and comorbidities, obesity is a major risk factor[8–11]. The association between obesity and severity has already been described for other viral disease. Influenza is the prototypical example[1,12,13], although there are notable exceptions[14–16]. The pathophysiological mechanisms underlying the greater severity of COVID-19 in obese individuals have yet to be identified but presumably include (i) factors that are common to all infections, and (ii) factors that are more specific to infections by SARS-CoV-2. Due to the variety of impairments associated with obesity (which go far beyond simply having excess adipose tissue (AT)), various mechanisms have been considered[17–19]. They range from the mechanical pressure on the lungs caused by visceral obesity[20–22] to metabolic disorders affecting first the AT and then systemic immune responses[23] and/or epithelial cells[24]. Obesity is also associated with activation of the renin-angiotensin system (notably in AT)[25], hypercoagulability[26], and comorbidities (e.g. type II diabetes, hypertension, and dyslipidemia) that are major risk factors for severe forms of COVID-19.

A major underlying mechanism is the systemic, low-grade inflammation associated with obesity that affects local immune responses (first in AT and then in other tissues) and subsequently induces systemic inflammation[27]. This chronic inflammation might contribute to the cytokine storm associated with severe forms of COVID-19[28]. Indeed, AT is central to this inflammatory profile because its endocrine activity shifts from anti-inflammatory cytokine production in lean individuals towards pro-inflammatory cytokine production (e.g. IL-6, TNF-α, and MCP-1) in obese individuals. Elevated levels of these three cytokines have been identified in severe forms of COVID-19[29]. Among the pro-inflammatory factors specifically expressed more in obese individuals, the pro-inflammatory hormone leptin is thought to be involved in COVID-19[30]. On the systemic level, obesity is also associated with free fatty acid overload[31], hyperglycemia[32], and high levels of oxidative stress[33], all of which contribute to sustained inflammation. Activation of the complement system[34] and disruption of vitamin D availability[35] might also contribute to pro-inflammatory responses.

Furthermore, it was recently shown that CD8 T cells in AT have a direct anti-infective activity[36], and that obesity has an impact on these T cell responses[37]. Lastly, the documented expression of angiotensin-converting enzyme 2 (ACE2) in AT[38] has also raised the question as to whether or not AT can be directly targeted by SARS-CoV-2 and contributes to the pathophysiology of COVID-19[39,40] by directly impacting the AT's metabolic and immune properties. This hypothesis is consistent with the notion that the COVID-19 is a multi-organ pathology that can also directly affect AT functions. Direct infection by SARS-CoV-2 might alter the AT's metabolic and immune functions and thus favor the development of excessive, pro-inflammatory responses. SARS-CoV-2 infection might be even more harmful in obese individuals because it might synergize with pre-existing quantitative and qualitative defects in AT. The detection of SARS-CoV-2 in AT also prompts the question of whether this tissue is a viral reservoir and whether it contributes to "long COVID-19".

Given that AT is the pathophysiological core feature of obesity, the primary objective of the present work was to determine the direct and indirect impacts of SARS-CoV-2 infection on both subcutaneous AT (SCAT) and visceral AT (VAT). Both tissues exhibit specific metabolic and immune features[41–43]. Our preliminary goal was to confirm the presence of SARS-CoV-2 in AT. Although the literature data on this topic are scarce, recent publications demonstrate the infection of adipocytes either in vitro assay or on autopsic samples for patients having died from COVID[44–48]. The existence of AT infection in the context of mild infection has not been addressed. We studied a model of experimental SARS-CoV-2 infection in cynomolgus macaques[49]: these non-human primates (NHPs) showed transient radiographic lung abnormalities (within 2–5 dpi), evidence of T cell activation, mild lymphopenia, and neutrophilia. We screened for SARS-CoV-2 in abdominal SCAT, VAT, and epicardial AT (EpAT), since the latter has been identified as a site of major inflammation[50]. We also performed a standard analysis of the immune compartments of SCAT and VAT, with a focus on CD8 T cells and on T cell markers (such as CD69, a marker of activation and residency in tissue[51,52], and PD-1[53], a marker associated with chronic infection and exhaustion) that are strongly expressed in AT. Although we studied a small number of animals, we detected SARS-CoV-2 in SCAT but not in VAT or EpAT. Interestingly, we also observed severe changes in the proportions of CD69+ and PD-1+ cells in both SCAT and VAT - suggesting that the systemic inflammatory response also affected AT.

## Results

**The animal cohort.** A total of ten cynomolgus macaques were included. The infected group comprised five females, which were infected by combined intranasal and intratracheal inoculation of SARS-CoV-2 (the strain first isolated in France: BetaCoV/France/IDF/0372/2020 SARS-CoV-2). The median [interquartile range (IQR)] age was 7.6 years [7.6; 7.7], and the median weight was 7.6 kg [7.6; 7.7]. The uninfected (control) group also comprised five females (median age: 5.4 years [5.4; 7.0]; median weight: 6.7 kg [3.8; 7.1]). The infected and uninfected groups did not differ significantly with regard to bodyweight but differed in age ($p = 0.0079$). The infection and follow-up protocols are summarized in Fig. 1a. Macaques were inoculated on day 0 with $1.0 \times 10^6$ PFU of SARS-CoV-2. Nasopharyngeal and tracheal swabs were collected daily between 0 and 7 dpi. Macaques were euthanized at 7 dpi, and selected fluids (nasopharyngeal and tracheal swabs) and tissues (SCAT, VAT, and EpAT) were collected at sacrifice for subsequent viral load, molecular and immunological assays.

We first confirmed that SARS-CoV-2 infection had occurred by analyzing the viral loads in nasopharyngeal swabs, tracheal swabs, and bronchoalveolar fluids. As mentioned above, the virus was detected in all five infected macaques and at all three respiratory tract sites (Fig. 1b). The viral load in nasopharyngeal swabs peaked between 1 and 6 dpi, with a median peak value of 8.1 $\log_{10}$ copies/mL at 3 dpi. The viral loads then fell progressively but were above the assay's limit of detection throughout the course of the study. In tracheal swabs, the viral load peaked at 2 dpi, with a median peak value of 7.5 $\log_{10}$ copies/mL at 3 dpi. The viral load fell rapidly between 3 and 5 dpi but we observed a viral rebound shortly before the time of euthanasia, with a median value of 4.8 $\log_{10}$ copies/mL at 6 dpi. Mild clinical signs and weight loss (median (range): 2.5% (1.4–4.2) of bodyweight) were observed. Biological monitoring of the inflammatory response in the blood revealed a trend towards an increase in MCP-1 at 2 dpi. No change in the concentrations of IL-6, TNF-α, IL-1β, cytokines known to be produced by AT in inflammatory contexts were detected between d0 and 7 dpi (Fig. 1c). We also measured the impact of the infection on metabolic parameters, such as adiponectin and leptin, predominant cytokines produced by AT, and insulin as AT dysfunction has also been associated to the insulin pathway (Fig. 1d). At 7 dpi, a trend towards a decreased leptin level associated to increased adiponectin concentration in the plasma, suggesting AT was indeed impacted during SARS-CoV-2 infection. Finally, we also observed a trend towards increased insulin level at 7 dpi. Collectively, we observed both metabolic and inflammatory changes associated with the SARS-CoV-2 infection induced in the cynomolgus macaque model, in accordance with previous publications[49].

**a**

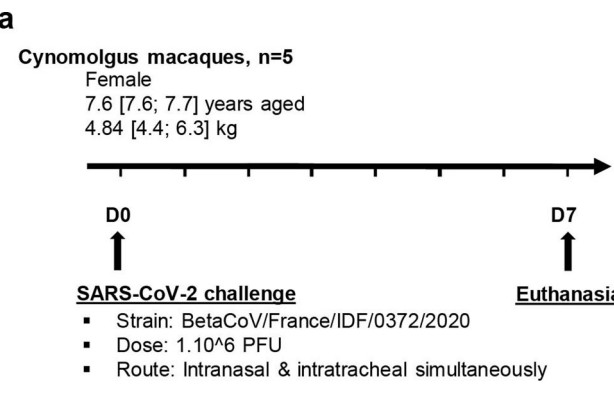

**b**

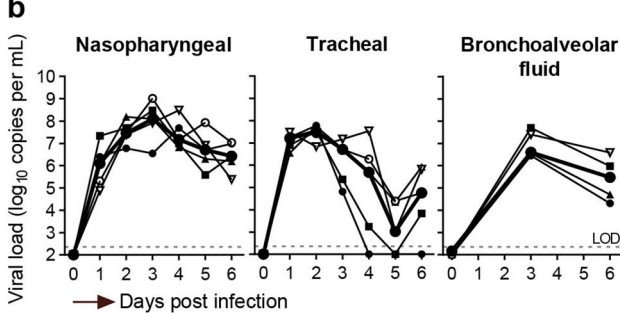

**c**

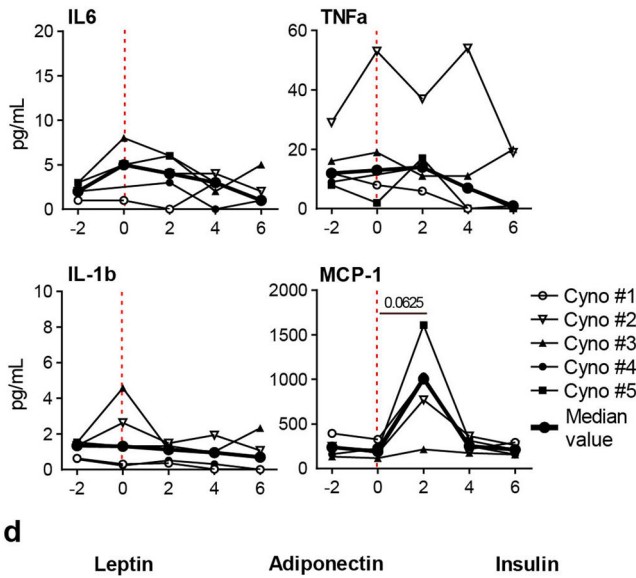

**d**

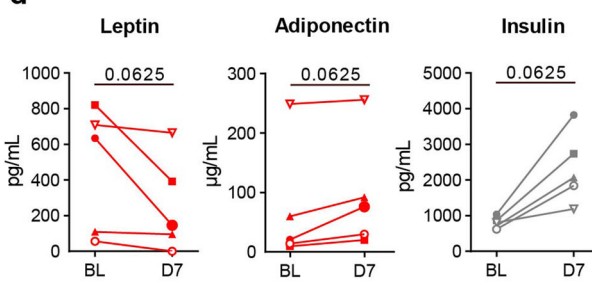

**Fig. 1 Study design and determination of the viral load in the respiratory tract. a** Experimental design, with the intratracheal and intranasal inoculation of the BetaCoV/France/IDF/0372/2020 A viral strain in NHPs (cynomolgus macaques). **b** Changes over time in the SARS-CoV-2 load in nasopharyngeal swabs (left panel), tracheal swabs (middle panel), and bronchoalveolar fluids (right panel) for the five infected NHPs tested. Viral loads were determined in an RT-PCR assay. The estimated limit of detection (represented by the dashed horizontal line) was 2.3 $\log_{10}$ copies RNA/mL. **c** Dynamic of cytokine plasmatic concentrations during SARS-CoV-2 infection: IL-6 (top left), TNF-α (top right), IL-1β (bottom left), MCP-1 (bottom right). **d** Changes in plasma leptin concentrations (left panel, adiponectin (middle panel) and insulin (right panel) at baseline (BL) and D7 after infection. Each NHP is represented by a single symbol ($n = 5$ independent animals), and the median viral loads are shown as a bold line. The values were compared in a Friedman test with Dunn's multiple comparison post-test. Statistically significant differences are indicated as follows: *$p < 0.05$.

load of 1.82 $\log_{10}$ copies/µg RNA [0.90; 2.23] (Fig. 2a). The virus was never detected in VAT or EpAT but was detected consistently in nasopharyngeal swabs (five out of five animals) and less consistently in tracheal swabs or rectal fluids (3 out of 5) (Fig. 2b). The median viral loads (and IQR) in nasopharyngeal, tracheal and rectal fluid samples were 6.2 [5.77; 7.42], 3.6 [0; 4.82] and 3.4 [0; 5.85] $\log_{10}$/ ml, respectively whereas the median viral load in SCAT lysate was 3.5 $\log_{10}$/ml [2.57; 3.93]. It is noteworthy that when the virus could not be detected in an animal's SCAT, it was also not detected in tracheal swabs or rectal fluid at the same timepoint. In contrast, one animal did not have a detectable viral load in tracheal and rectal fluids but still had virus in the AT. An important question is the long-term persistence of the virus in the SCAT. We had the opportunity to check the viral load in five male animals at 43 dpi. No virus was detected at this later timepoint suggesting a transient infection of AT.

**Higher expression of ACE2 in SCAT than in VAT.** The presence of virus in SCAT has also been demonstrated in the setting of influenza[54], suggesting that this tissue has specific metabolic, immune and/or vascular properties. However, SARS-CoV-2's infectious cycle is highly dependent on the expression of various entry receptors, such as ACE2 and TMPRSS2. We therefore measured the mRNA expression of ACE2 in SCAT and VAT samples from seven cynomolgus macaques (five infected with SARS-CoV-2, 7 dpi: one animal with a chronic simian immunodeficiency virus infection, and one uninfected animal) (Fig. 2c). The small size of the EpAT samples prevented us from assaying them for ACE2 expression. ACE2 mRNA expression was found to be significantly (7-fold) higher in SCAT than in VAT whereas TMPRSS2 mRNA expression was not detected in all samples. Overall, these findings provide a rationale for the infection of SCAT (but not VAT or EpAT) during SARS-CoV-2 infection, although high heterogeneity is observed among SCAT samples. Due to the uncertainties related to mRNA expression as ACE2 expression is also highly regulated by age, sex, metabolic context, AT composition or SARS-CoV-2 infection[45,55,56], we also confirmed at the protein level the expression of ACE2 in AT by immunochemistry. For technical reason (no ACE2 antibody available for the cynomolgus model), this experiment was performed on human AT collected from male and female donors. The median of age was 54.1 [38.1–69.6] years and the body mass index was 33 [27.5–40.6]. We detected ACE2 in the cytoplasm of the adipocytes, below the plasmic membrane and close to the nucleus (Fig. 2d). Interestingly, contrary to the mRNA data, we observed a similar proportion of adipocytes expressing ACE2 in SCAT and VAT, 10–15% of adipocytes expressed ACE2. (Fig. 2e). This discrepancy between mRNA

**Detection of SARS-CoV-2 in subcutaneous AT.** We next established whether or not SARS-CoV-2 could be detected in AT. Using an RT-PCR assay, we measured the infected animals' viral loads in SCAT, VAT, EpAT, and nasopharyngeal, tracheal and rectal fluids 7 dpi (i.e. after euthanasia). SARS-CoV-2 was detected in the SCAT from 4 of the 5 infected animals (80%), with a median viral

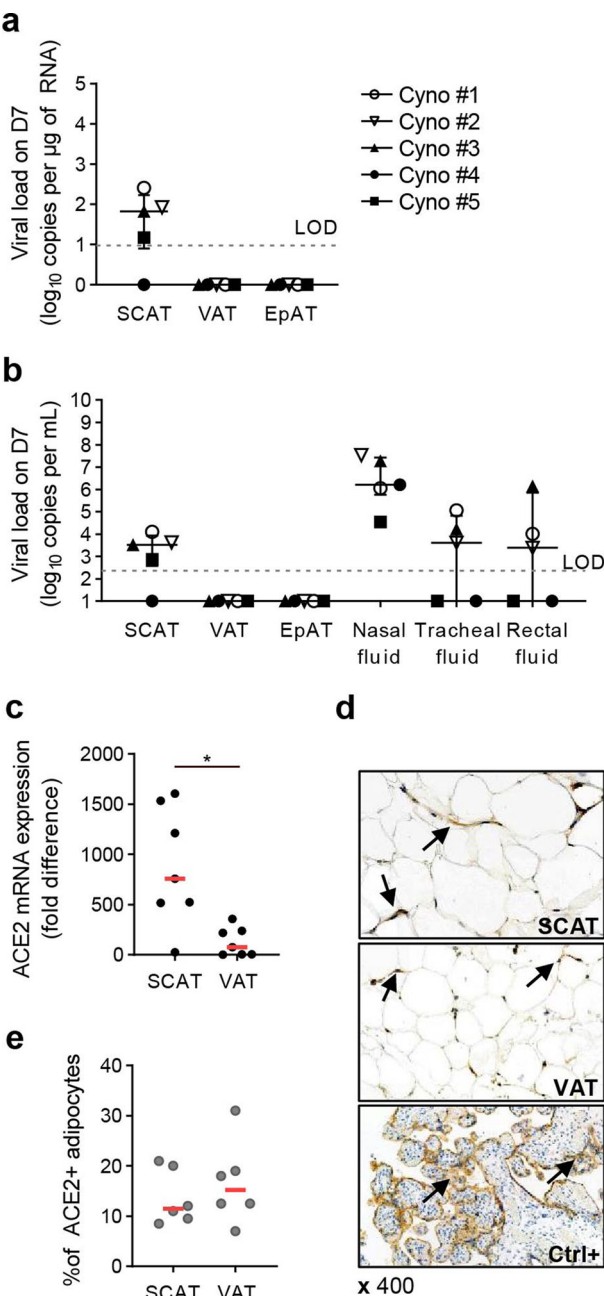

**Fig. 2 Detection of SARS-Cov-2 in SCAT, and higher levels of ACE2 mRNA expression in SCAT than in VAT.** Characterization of the AT during SARS-CoV-2 infection in cynomolgus macaques, with quantification of SARS-CoV-2 RNA and the expression of the cell-entry receptor angiotensin-converting enzyme 2 (ACE2). **a** Quantification of SARS-Cov-2 RNA in SCAT, VAT, and EpAT and IQR. The estimated limit of detection (represented by the dashed horizontal line) was 9.5 copies/µg of RNA. The median and errors bar shown as interquartile [IQR] are indicated. **b** The viral loads in SCAT, VAT and EpAT lysate, and the nasal, tracheal, and rectal fluids at euthanasia and IQR (7 dpi). The estimated limit of detection (represented by the dashed horizontal line) was 2.3 $\log_{10}$ copies RNA/mL. The median and errors bar shown as interquartile [IQR] are indicated. Viral loads were determined in an RT-PCR assay. Each NHP is represented by a single symbol ($n = 5$). **c** Semi-quantitative measurement of ACE2 mRNA expression in SCAT and VAT, normalized against the *PPIA* housekeeping gene. Data were generated from SCAT and VAT collected from 7 NHPs. Each animal is represented by a pair of linked points. The graph shows the fold-difference in ACE2 expression. The values were compared in a paired nonparametric Wilcoxon test. Statistically significant differences are indicated as follows: *$p < 0.05$, **$p < 0.01$, ****$p < 0.0001$. **d** Immunohistochemical analysis of ACE-2 expression in human AT. ACE-2 expression was analyzed in SCAT (upper), VAT (middle) and placenta as positive control (lower). Brown staining for ACE2, blue staining for DAPI. **e** Proportion of adipocytes expressing ACE-2 at the protein levels in human adipose tissue samples from non-COVID patients. Each patient is represented by a symbol ($n = 6$ independent patients, $n = 6$ SCAT and $n = 6$ VAT). The values were compared in a paired nonparametric Wilcoxon test.

and protein data may reflect the different composition of AT in SCAT and VAT: due to the higher proportion of stromal vascular cells in VAT, the expression of ACE2 mRNA in whole VAT may appear lower than what is observed in SCAT. Further investigation will be required to understand the mechanisms regulating ACE2 but we confirm the expression of ACE2 in AT, both in cynomolgus macaques and humans.

**SARS-CoV-2 infection did not affect the proportions of CD4 and CD8 T cells in SCAT or VAT.** The immune compartment of AT is critically involved in regulation of the tissue's metabolic activity and the development of low-grade inflammation. Furthermore, AT is also a site where memory CD8 T cells with anti-infective immune functions accumulate. Since the direct infection of AT might induce major local immune activation and disrupt the tissue's endocrine functions, we next used flow cytometry to evaluate the impact of SARS-CoV-2 infection on the immune compartment in fresh SCAT and VAT samples. As a control, we

reanalyzed data from earlier studies of fresh samples from animals not infected with SARS-CoV-2. As a prerequisite, we compare the number of stromal vascular cells collected following AT dissociation. No significant difference in SVF cell numbers was observed in the SCAT and VAT collected from SARS-CoV-2 infected animals compared to control animals (Fig. 3a). Our analyses focused on the T cell fractions, and the gating strategy is shown in Supplementary Fig. 1. We first examined the proportion of CD45[+] cells in the AT's stromal vascular fraction (SVF), as a marker of change in AT SVF composition. As shown in Fig. 3b, the infected and non-infected macaque groups did not differ significantly in terms of the proportion of CD45[+] cells in the SCAT or the VAT. The median [IQR] proportion of CD45[+] cells in SCAT SVF was 28.6% [24.3; 36.5] in the infected group and 19.3% [12.1; 37.8] in the control group. In VAT SVF, the median [IQR] proportion was 43.1% [37.2; 61.9] in the infected group and 41.9% [23.6; 55.0] in the control group.

We next determined the effect of SARS-CoV-2 infection on CD4[+] and CD8[+] distribution among CD45[+] cells (Fig. 3c); AT CD4 and CD8 T cells have antiviral properties, are involved in the metabolic regulation of AT[52,57], and are severely reduced in the blood of patients with severe forms of COVID-19. There was no significant difference between infected and non-infected macaques in the proportion of CD4 T cells in SCAT or in VAT. The same was true for the proportion of CD8 cells.

Overall, we did not observe major quantitative changes in the proportion of CD45[+] cells or CD4 and CD8 T cell fractions in SCAT and VAT at 7 dpi.

**SARS-CoV-2 infection is associated with a drastic reduction in CD69 expression on T cells from SCAT and VAT.** Although the infection-associated changes in AT T cell proportions were limited, we also analyzed the expression of standard T cell markers (such as PD-1 and CD69) and other markers of cell activation

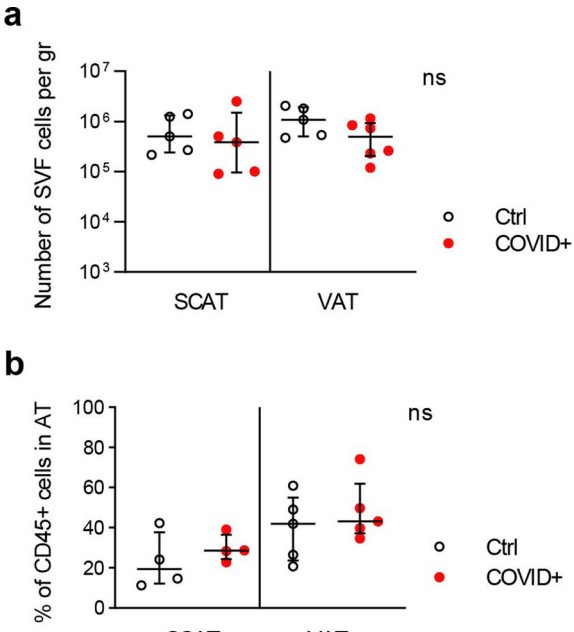

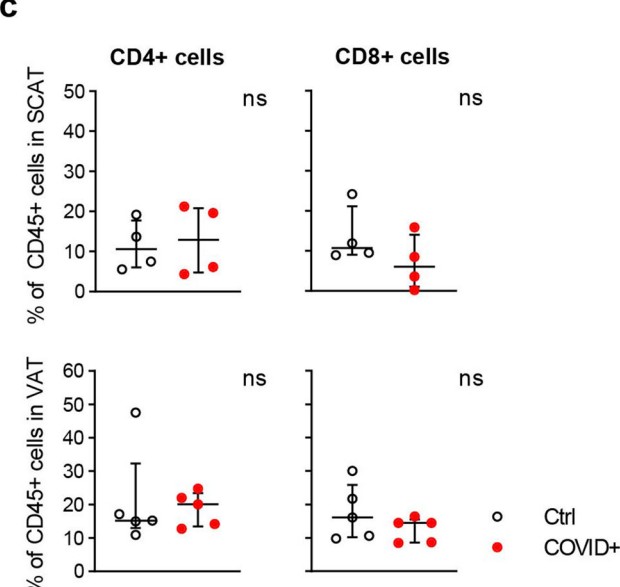

**Fig. 3 Slight changes in the AT T cell compartment seven days after SARS-CoV-2 infection vs. controls.** Comparison of the number of SVF cells per gram of AT, the proportions of CD45+ cells, CD8 T cells and CD4 T cells in d7 SARS-CoV-2 animals vs. age-matched, sex-matched, weight-matched control (uninfected) cynomolgus macaques. **a** The number of SVF cells per gram of AT. **b** CD45+ cells as a proportion of live cells in the SVF from SCAT and VAT, as determined by flow cytometry. **c** The proportions of CD4+ and CD8+ cells among CD45+ SVF cells from SCAT and VAT. The graphs show data from uninfected NHPs (black open circles: $n = 4$ for SCAT, $n = 5$ for VAT) and from SARS-CoV-2-infected NHPs (open circles with a red point: $n = 4$ for SCAT, $n = 5$ for VAT). The median [IQR] proportion of CD4+ and CD8+ cells in SCAT SVF was 12.9% [4.8; 20.8] and 6.0% [1.0; 14.1] respectively in the infected group and 10.6% [6.0; 17.8] and 10.8% [9.1; 21.1] respectively in the control group. In VAT SVF, the median [IQR] proportion of CD4+ and CD8+ cells was 20.1% [13.5; 23.4] and 14.5% [8.6; 15.5] respectively in the infected group and 15.2% [13.0; 32.3] and 16.1% [10.2; 25.9] respectively in the control group. Each symbol represents a distinct animal. The median and errors bar shown as interquartile [IQR] are indicated. Statistically significant differences (as determined in a nonparametric, unpaired Mann–Whitney test) are indicated as follows: $*p < 0.05$, $**p < 0.01$, $****p < 0.0001$.

and proliferation (Figs. 4a and 4b). In non-infected macaques, the SCAT and VAT contained similar proportions of CD69+ T cells (ranging from ~35% to ~45%); this was true for both CD4 and CD8 subsets. In the blood, the proportion of CD69+ T cells was low (Supplementary Fig. 2)[53]. In infected animals, however, we observed a collapse in the proportion of CD69+ T cells in both SCAT and VAT at 7 dpi. The CD4 and CD8 T cell subsets were both affected. The median [IQR] proportion of CD69+ CD4 T cells in SCAT SVF was 2.1% [0.8; 3.2] in infected animals and 35.2% [30.2; 47.9] in controls ($p = 0.0286$); the median [IQR] proportion in VAT SVF was 1.8% [0.8; 4.7] in infected animals and 43.7% [43.7; 60.4] in controls ($p = 0.0079$). The median proportion of CD69+ CD8 T cells in SCAT SVF was 1.6% [0.4; 3.7] in infected animals and 35.7% [30.1; 40.8] in controls ($p = 0.0286$); the median [IQR] proportion in VAT SVF was 0.8% [0.1; 3.3] in infected animals and 32.4% [22.5; 37.5] in controls ($p = 0.0079$). In the blood, the proportion of CD69+ T cells remained low, but exhibited significant and opposite changes

among CD4 and CD8 T cells: the proportion of CD69+ cells being higher among CD4 T cells and lower among CD8 T cells in the blood of SARS-Cov-2 infected animals (Supplementary Fig. 2). To note, at 7 dpi, there is no more trace of T cell lymphopenia, as observed at 2 dpi (Supplementary Fig. 3). We also evaluated the proportions of PD-1+ T cells in AT: PD-1 (an activation and/or exhaustion marker on T cells) is usually expressed strongly in AT. Indeed, high proportions of PD-1+ CD4 and CD8 cells were observed in both SCAT and VAT from control (uninfected) macaques. In SARS-CoV-2 infected animals, a higher proportion of PD-1+ cells was observed in VAT but not SCAT (Fig. 4b). This difference was observed for both CD4 and CD8 T cell subsets. The median proportion of PD-1+ CD4 T cells in VAT SVF was higher in infected animals than in controls ($p = 0.0151$), 51.8% [38.1; 59.9] and 34.6% [20.3; 37.1] respectively. In the CD8 T cell compartment, median proportion of PD-1+ cells was 37.7% [30.7; 47.3] in infected animals and 8.9% [6.6; 21.4] in controls ($p = 0.0079$).

We also measured the expression of HLA-DR (a standard marker of T-cell activation) and Ki67 (a proliferation marker), neither of which are highly expressed by AT T cells under baseline conditions[53]. As expected, the proportions of HLA-DR+ CD4 and CD8 T cells were similar in SCAT and VAT from uninfected animals, with values of between 5 and 10% (Fig. 4c). The proportion of HLA-DR+ CD4 T cells in the AT was lower in infected macaques than in uninfected macaques. No changes in the CD8 T cell compartment were detected. In SCAT, the median proportion of HLA-DR+ CD4 T cells was 2.3% [1.0; 3.8] in infected animals and 8.2% [5.7; 16.2] in control animals ($p = 0.0286$). In VAT, the median proportion was 3.4% [1.9; 5.0] in infected animals vs. 8.1% [5.1; 12.9] in control animals ($p = 0.0317$). In infected animals, the differences between SCAT and VAT where not significant. Lastly, the AT T cells' proliferative potential was assessed by evaluating Ki67 expression (a marker of non-resting cells).

There were no significant differences in the proportions of Ki67+ CD4 or CD8 cells between infected and uninfected macaques or between SCAT and VAT samples.

In summary, our phenotyping of AT T cells revealed contrasting alterations 7 days after SARS-CoV-2 infection. The proportions of CD69+ T cells in SCAT and VAT were much

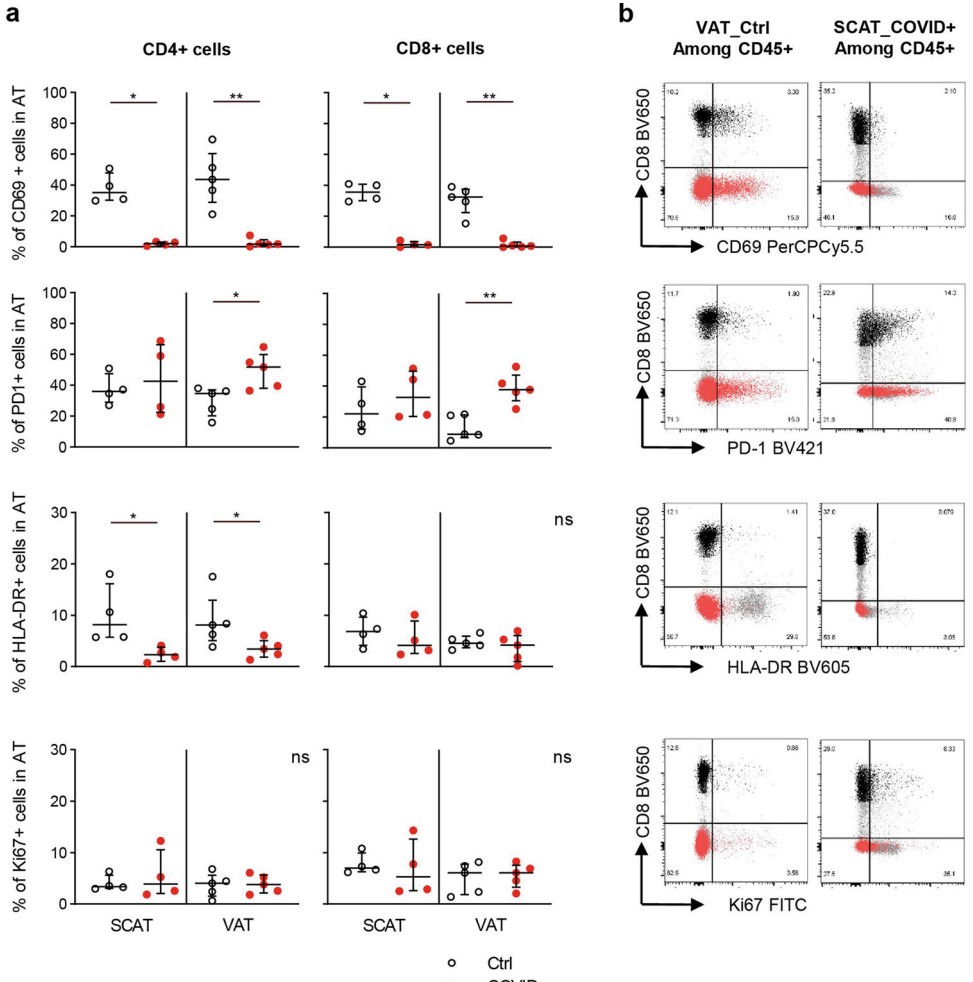

**Fig. 4 Phenotyping of CD4[+] and CD8[+] cells in SCAT and VAT.** Comparison of the phenotypic features of CD8 and CD4 T cells on day 7 in SARS-CoV-2-infected animals vs. age-matched, sex-matched, weight-matched control (uninfected) cynomolgus macaques, using flow cytometry. **a** Proportions of CD69[+], PD-1[+], Ki67[+], and HLA-DR[+] CD4 T cells (left graph) and CD8 T cells (right graph) (median [IQR]). The median and errors bar shown as interquartile [IQR] are indicated. **b** Representative dot plots for CD69, PD-1, Ki67 and HLA-DR expression on CD4 (red dot) and CD8 T cell (black dot) subsets from SCAT and VAT, as determined by flow cytometry. The graphs show data from uninfected NHPs (black open circles: $n = 4$ for SCAT, $n = 5$ for VAT) and SARS-CoV-2 infected NHPs (open circles with a red point: $n = 4$ for SCAT, $n = 5$ for VAT). Each symbol represents a distinct animal. Statistically significant differences (as determined in a nonparametric, unpaired Mann–Whitney test) are indicated as follows: *$p < 0.05$, **$p < 0.01$, ****$p < 0.0001$.

lower in infected animals than in uninfected animals. We also observed tissue-specific changes (such as a higher proportion of PD-1[+] cells in the VAT only) and cell-subset specific changes (such as a lower proportion of HLA-DR[+] cells in the CD4 subset only). The data on the AT's immune activation status were also contrasting. The drastic reduction in CD69 expression observed in both SCAT and VAT suggests that even areas of AT not directly infected by SARS-CoV-2 might be altered by the systemic inflammatory responses developed during the infection.

## Discussion

The association between obesity and the severity of COVID-19 has raised questions about the underlying mechanisms. Obesity is commonly associated with comorbidities and low-grade local and systemic inflammation. The role of these parameters need to be addressed, and AT has a key role in the development of these inflammatory phenomena. Given that AT expresses ACE2, we hypothesized that the virus could directly infect AT and could subsequently alter the tissue's metabolic and immune functions.

In a macaque model of SARS-CoV-2 infection, we detected the virus at 7 dpi in 4 of the 5 SCAT samples tested. In contrast,

SARS-CoV-2 was never detected in VAT or EpAT. The specific presence of SARS-CoV-2 in SCAT is in line with the literature data for influenza virus[54]; SCAT's specific metabolic, immune and vascular properties might make it more susceptible than other sites[58]. The greater expression of ACE2 in SCAT than in VAT might explain this selective infection. In the literature, the data on ACE2 expression in AT are scarce and discordant. One study found that ACE2 mRNA expression is higher in AT than in lung[59]. The impact of obesity on ACE2 mRNA expression remains unclear; some publications have suggested that ACE2 expression is regulated in different metabolic contexts[60–62], whereas another did not detect any differences[63]. Taken as a whole, these data suggest that the regulation of the receptor's expression in AT is complex and depending on the type of AT considered[61], the sex[60], the exposure to metabolism-modifying medications[64] and the composition of the AT[56]. It is noteworthy that downregulation of ACE2 expression in the lung is observed in COVID-19, following internalization of the receptor with the virus[65]. A recent publication describes an increase in ACE2 transcription levels in AT following COVID-19 infection[45] in contrast to previous publication describing a downregulation of

ACE2 following SARS-CoV binding in the lung[55]. Additionally, influenza infection and exposure to type I interferon increase ACE2 expression in the lung[66]. There is no consensus on ACE2 expression levels in SCAT vs. VAT; it has variously been reported that the level is similar in both ATs[59] or is higher in VAT than in SCAT[64]. In the present study, we observed higher mRNA expression of ACE2 in SCAT than in VAT in 7 samples from both SARS-CoV-2-infected animals and uninfected animals. The higher expression of ACE2 may contribute to the preferential infection of SCAT. This discrepancy with regard to previous studies may reflect differences in the model used (i.e. humans vs. NHPs), the metabolic context (obese vs. nonobese animals), the severity of the viral disease (lethal vs. mild), and the timepoint of the analysis (autopsy vs. 7 dpi). It may also reflect the intrinsic limitation of mRNA detection in whole AT. Higher expression of ACE2 mRNA in SCAT may not reflect higher expression on a per cell basis, but rather a higher relative proportion of adipocyte among the complex cellular network that is AT and the difference in composition between SCAT versus VAT[56]. We thus evaluated the expression of ACE2 at the protein level by immunohistochemistry on human samples. This different method did not reveal any difference in the proportion of adipocytes expressing ACE2 in both AT. Nevertheless, in a model of early-stage infection (i.e. nonsevere disease), we demonstrated that AT is effectively targeted by the SARS-CoV-2. This observation is in accordance with the report showing the presence of SARS-CoV-2 in autopsy samples of SCAT from patients who died of COVID-19. However, our results also suggest that SCAT is infected in the context of mild COVID-19. The mechanisms underlying the preferential infection of SCAT by SARS-CoV-2 remain to be determined. This finding also raises questions with regard to (i) the duration of SARS-CoV-2's persistence in AT, (ii) possible spreading of the virus to all AT sites over the course of severe COVID-19, and (iii) the local impact of SARS-CoV-2 on SCAT's metabolic and immune properties. Concerning the persistence of the infection in AT, we had the opportunity to measure the viral load at 43 dpi in a cohort of male animals. In that context, no virus was detected, suggesting no long-term sequestration of the virus in AT in this model of mild COVID. The absence of VAT infection may also reflect the transient lung infection seen in mild COVID, that may reduce the possibility for large spread. Alternately, one cannot exclude that VAT was infected but eliminated the virus prior 7 dpi. Indeed, VAT includes more hematopoietic and immune cells and may thus eradicate the virus more rapidly than the SCAT. Although the severity of COVID-19 has been linked to visceral adiposity[61], it is important to note that both SCAT and VAT might contribute to the metabolic alterations associated with obesity[67–69]. VAT is thought to be more metabolically active, whereas SCAT is predominantly a lipid buffering site that prevents lipotoxicity.

We next focused on the immune properties of AT in SARS-CoV-2-infected animals by studying the T cell compartment in both SCAT and VAT. The proportion of CD45+ cells did not significantly differ when comparing uninfected and infected animals at 7 dpi. The proportions of CD4 and CD8 T cells were similar in SCAT and in VAT. This result was surprising; although SCAT was infected by SARS-CoV-2, it did not show any signs of an overt immune response. A reduction in the proportions of both CD4 and CD8 cells might have indirectly indicated the recruitment/activation of the innate immune system, as has been seen for neutrophils and monocytes in the lungs during SARS-CoV-2 infection[70]. Conversely, an increase in the proportions of CD4 and/or CD8 cells would have indicated the induction of a T-cell-specific response because both CD4 and CD8 T cells are involved in the defense against different types of virus. Unfortunately, we were unable to provide absolute T cell counts for AT

from SARS-CoV-2-infected and control animals; this prevented us from determining whether or not T cell lymphopenia had developed in the AT (as has been described in the blood[71]) or whether T cell subsets had been recruited from the blood.

Even though the proportions of CD4 and CD8 T cells were unchanged, we observed a marked difference in the expression of CD69 on T cells collected from SCAT and from VAT. The proportions measured in uninfected animals were in line with the literature data but CD69 expression was dramatically lower in infected animals. CD69 is expressed on activated T cells and tissue-resident T cells. The expression of CD69 counteracts the expression of the sphingosine-1 phosphate receptor that ensures local T cell egress; hence, CD69 thus helps to retain T cells in tissues[72]. The loss of CD69 expression on AT T cells might therefore reflect (i) a major change in the migratory potential of AT T cells, leading to remobilization and a contribution to systemic hyperactivation, and/or (ii) functional alterations in AT-resident T cells caused by inappropriate activation[73]. Importantly, the expression of CD69 also modulate the metabolic activity of T cells[74], which could be especially relevant in the metabolic tissue that is AT. Taken as a whole, these results suggest that (i) CD69 expression at the surface of AT T cells changes dramatically, and (ii) the CD69+ AT T cell fraction was not replaced. The total proportions of CD8 and CD4 cells did not differ when comparing samples from infected and uninfected animals, and PD-1 expression was modulated in a different way in VAT (with a higher proportion of PD-1 expressing cells) relative to SCAT. These findings suggest that CD69 is specifically regulated in what would be a new pathophysiological mechanism in SARS-CoV-2 infection. The details of the mechanism and consequences of the loss of CD69 expression on AT in this specific context need to be evaluated further. However, the lower proportion of HLA-DR + CD4 T cells might be related to abnormal regulation of T cell activation in AT. It is tempting to speculate that the increase in CD69+ cells among CD4 T cells in the blood is a reflect of the remobilization of the AT CD4 T cells and may contribute to the restoration of the blood T cell count. However, as CD69 is a marker of T cell activation, further investigations will be required to evaluate the contribution of AT CD4 T cells to the blood CD4 T cell compartment.

Despite the persistence of equivalent CD4 and CD8 T cell proportions, the higher observed expression of PD-1 in VAT of infected animals but not SCAT emphasized the differences between the two AT's immune properties. This difference might result in a particular immune and/or metabolic microenvironment and might thus impact SARS-CoV-2 infection. This might seem counterintuitive because PD-1+ CD8 and CD4 T cells in blood or in cancer are usually considered to be exhausted. However, PD-1 is also a marker of cell activation, and its expression on T cells is associated with a lower level of glycolysis, preferential lipolysis, and greater fatty acid oxidation[75]. Furthermore, this metabolic pathway has been linked to greater antiviral activity for CD8 T cells in other infections (e.g. HIV infection[76]) and might reflect a beneficial adaptation to the AT environment, rather than an impaired response. The direct impact of PD-1 modulation in VAT at 7 dpi remains to be elucidated.

This work has obvious limitations. Firstly, we did not provide any insight on the type of cells that bear the virus in the SCAT. Indirectly, the identification of ACE2 protein preferentially on adipocytes clearly points towards adipocytes as crucial target of the infection. The literature in the field has now provided some important insight confirming the infection of adipocytes, both in vitro and ex vivo[45–48], although the infection of AT macrophages has also been described[46]. Secondly, the analyses are performed in a context of mild infection. This setting may explain some of the discrepancy observed compared to other publications

in terms of the immune or metabolic signature of the infection. However, it also confirms that the infection of AT is not a process that is strictly associated with the most severe form of infection. Interestingly, the change in the phenotypic of AT T cells is drastic, although not impacting the outcome of this acute infection. The long-term consequences of these defects will need to be further evaluated.

Overall, our work shows that SARS-CoV-2 infection impacts AT two levels: (i) through the direct infection of SCAT, and (ii) through bystander effects on both uninfected and infected AT. The post-infection inflammatory responses in the lung and then systemically are strong candidates for altering the AT's immune composition and thus its immune and metabolic properties - regardless of the presence of the virus per se at each site. Out results confirm the need to study the impact of SARS-CoV-2 on AT in obese and/or aged individuals and in men vs. women. It is tempting to speculate that the virus's metabolic and immune effects on AT will be exacerbated in obese individuals. The impact on the AT T cell compartment might be more severe, due to the higher proportion of pro-inflammatory T cells in AT in obese people prior to infection. The abnormalities observed in our NHP model strongly suggest that AT dysfunction has a major role in the pathogenesis of COVID-19 and warrants further investigation.

## Methods

**Animal models**. Cynomolgus macaques (*Macaca fascicularis*) were imported from AAALAC-certified breeding centers in Mauritius. All animals were housed at the IDMIT facility (CEA, Fontenay-aux-Roses, France) in BSL-2 and BSL-3 containment conditions, when necessary (animal facility authorization reference: D92-032-02, Hauts de Seine Prefecture, France). All studies were performed in compliance with the European Directive 2010/63/EU, French regulations, and the Standards for Human Care and Use of Laboratory Animals published by the US Office for Laboratory Animal Welfare (OLAW, assurance number #A5826-01). The experiments were approved by an institutional review board (*Comité d'Ethique en Expérimentation Animale du Commissariat à l'Energie Atomique et aux Energies Alternatives*, Fontenay-aux-Roses, France; reference: CEtEA #44)). The study was authorized by the French veterinary authorities (registration number: APA-FIS#24434-2020030216532863v1).

**Infection with SARS-CoV-2**. Five female cynomolgus macaques were challenged simultaneously via the intranasal and intratracheal routes with $1.0 \times 10^6$ PFU of primary isolated SARS-CoV-2 (strain: BetaCoV/France/IDF/0372/2020). Body-weight, body temperature, and clinical signs of infection were monitored daily. Blood cell counts were determined from EDTA-treated blood samples using a HMX A/L analyser (Beckman Coulter). Nasopharyngeal and tracheal swabs were collected daily until 7 dpi. Bronchoalveolar fluids were collected at dpi 3 and 6, and rectal fluids were collected at dpi 0, 2, 5, and 7. At 7 dpi, animals were humanely euthanized with 18.2 mg/kg of intravenous pentobarbital sodium during anesthesia with tiletamine (4 mg/kg) and zolazepam (4 mg/kg), and an autopsy was performed. At the time of euthanasia, abdominal SCAT, VAT and EpAT samples were collected and assayed for SARS-CoV-2 and/or immune and cellular characteristics. A second cohort of cynomolgus macaques including six male animals were challenged and euthanized at later timepoint (43 dpi), i.e. after the resolution of the infection.

**Control samples**. Samples from historical cohorts of cynomolgus macaques were used as controls. Stromal vascular cells were collected from the VAT and SCAT of animals of the same sex and equivalent weights. The control group was composed of SCAT and VAT samples from 4 to 5 female cynomolgus macaques.

**Cell isolation**. Stromal vascular fraction was isolated from fresh AT samples. When necessary, the AT was devascularized before dissociation. AT was rinsed in Dulbecco's Modified Eagle's Medium (DMEM) (Lonza, Basel, Switzerland) with 5% fetal bovine serum (FBS), weighed, and cut into pieces of 2–3 mm (to optimize enzymatic digestion). The pieces were digested in a solution of type VIII collagenase from *Clostridium histolyticum* (0.33 mg/mL in DMEM with 5% FBS; Sigma-Aldrich, St. Louis, MO, USA). Enzymatic digestion was performed for 30 min at 37 °C with constant stirring. This was followed by mechanical dissociation by repeated suction-expulsion of the suspension with a 10 mL syringe. The AT suspension was filtered through a cell strainer (pore size: 100 μm; Corning, New York, NY, USA) and centrifuged at 330 g for 8 min at room temperature. The upper phase of the supernatant was discarded, and the lower phase (comprising the SVF cells) was centrifuged at 660 g for 8 min at room temperature. The supernatant was discarded, and the pellet containing the SVF was filtered through a cell strainer (pore size: 100 μm) and resuspended in DMEM with 5% FBS. Trypan blue-treated

cell suspensions were then counted under a microscope using a Malassez cell (C-chip, NanoEntek, Seoul, Korea). The SVF was either directly stained for flow cytometry analysis or cryopreserved in 90% FBS/10% DMSO.

**Cell staining and flow cytometry**. Fresh samples were stained after the incubation of SVF with 1:10 diluted Fc block reagent (BD Biosciences, Franklin Lakes, NJ, USA) for 15 min at 4 °C. Next, cells were incubated for 15 min at 4 °C with fluorescence-labelled antibodies against the following surface markers: 1:20 diluted CD45 (BV510, D058-1283), CD8 (BV650, RPA-T8), CD4 (BV711, L200), CD3 (BV786, SP34-2), CD69 (FN50, PercPCy5.5); 1:200 diluted HLA-DR (BV605, G46-6), from BD Biosciences; and 1:40 diluted PD-1 (BV421, EH12-2H7) from Biolegend (San Diego, CA, USA). Next, cells were incubated for 15 min at 4 °C with 1:2000 diluted amine-reactive blue dye (Live/Dead Fixable, Life Technologies, Carlsbad, CA, USA). After this incubation, cells were washed in PBS 1X and permeabilized–fixed using an Intracellular Fixation and Permeabilization Buffer Set (eBiosciences Scientific, Waltham, MA, USA) for 30 min at 4 °C. The cells were washed in PBS 1X, incubated with 1:10 diluted Fc block reagent (BD Biosciences) for 15 min at 4 °C, and then incubated with 1:20 diluted Ki67 (FITC, B56, from BD Biosciences) for 30 min at 4 °C. Lastly, the cells were washed in PBS 1X. Data were acquired with an LSR Fortessa flow cytometer (BD Biosciences) and analyzed using FlowJo software (version 10.6.2, FlowJo LLC, Ashland, OR, USA).

**Quantification of the viral load in NHP samples**. All specimens (whether nasopharyngeal, tracheal, or rectal) were collected with swabs (Viral Transport Medium, DSR-052-01, CDC, Atlanta, GA, USA) and stored at between 2 °C and 8 °C until analysis with a standard concentration range of plasmid containing an *rdrp* gene fragment including the RdRp-IP4 RT-PCR target sequence. AT samples were stored at -80 °C until analysis. Fragments of 50 mg of frozen SCAT and VAT samples were lysed in NucleoZOL (Machery-Nagel, Duren, Germany) using a Precellys device (Bertin Technology, Montigny-le-Bretonneux, France). RNA isolation and reverse transcription were performed according to the manufacturers' instructions (NucleoSpin™ RNA Core Kit (Macherey-Nagel). The protocol for detecting SARS-CoV-2 is available on the World Health Organization's website (https://www.who.int/docs/default-source/coronaviruse/real-time-rt-pcr-assays-for-the-detection-of-sars-cov-2-institut-pasteur-paris.pdf).

**Quantitative RT-PCR**. Fragments of 100–200 μg of frozen SCAT and VAT samples from seven cynomolgus macaques were dissociated in Qiazol (Qiagen, Hilden, Germany) using stainless steel beads in the tissue-lyser system (Qiagen). The RNA-containing aqueous phase was collected after treatment with chloroform. Total RNA from the aqueous phase was precipitated with ethanol. RNA isolation and reverse transcription were performed according to the manufacturers' instructions (NucleoSpin™ RNA Plus Kit (Macherey-Nagel) and the enhanced RT-PCR Kit (Sigma-Aldrich), respectively. ACE2 mRNA expression was determined using a standard reverse-transcription PCR assay and commercially available TaqMan probes (Applied Biosystems, Foster City, CA, United States) on the CFX96 thermocycler (Bio-Rad, Hercules, CA, United States). All results were normalized against expression of the *PPIA* gene. The *18 S* rRNA gene was used as second housekeeping gene. The TaqMan probes for ACE2 (*Mf01085327_m1*), *TMPRSS2* (*Mf02802837_m1*), *adiponectin* (*ADIPOQ Mf02788052_m1*), *leptin* (*LEP Mf02788316_m1*), *PPIA* (*Mf04932064_gH*), and *18 S* (*Hs99999901_s1*) were used.

**Protein detection**. Cytokines were quantified in EDTA-treated plasma using the NHP cytokine Milliplex (Merck Millipore, Darmstadt, Germany) for IL-6, TNF-α, IL-1β and MCP-1 (also known as CCL2); the NHP metabolic Milliplex panel for leptin and insulin; and the Human Adipokine Milliplex for adiponectin. Analysis was performed on a Bioplex 200 analyser (Bio-Rad) according to manufacturer's instructions.

**Microscopy and immunochemistry**. Samples of SCAT and VAT (and placenta as control) were fixed in 4% buffered formalin and embedded in paraffin. Sections (thickness: 3 microns) were immunostained for ACE2 using unconjugated recombinant rabbit monoclonal antibody directed against human ACE2 (clone SN0754) (Thermofisher scientific, Waltham, Massachusetts, USA) followed by incubation with Goat anti-Rabbit IgG (H + L) Highly Cross-Adsorbed Secondary Antibody. Nuclei were stained with DAPI. The merged image is shown at the magnitude of 400.

**Human sample**. SCAT and VAT samples from six patients who provided their written, informed consent to participation were selected for ACE2 protein detection. The study protocol was approved by the regional investigational review board (Comité de Protection des Personnes Ile-de-France VII, PP12-021, Paris, France). The study includes three female and three male donors. The median of age is 54.1 [38.1–69.6] and the median of body mass index is 33 [27.5–40.6].

**Statistics and reproducibility**. The size of the group was defined based on statistical and ethical parameters. Groups of 5 to 6 NHPs were studied for each group. No biological replicate was performed for the in vivo infection of NHPs. Technical

replicates were performed for RT-PCR, protein detection, viral detection and microscopy. All statistical analyses were performed using GraphPad Prism software (version 8, GraphPad Software Inc., San Diego, CA, USA). Continuous variables were described as the median [IQR]. Groups were compared using a nonparametric, unpaired Mann–Whitney two-sided test or a paired Wilcoxon two-sided test. Comparison of values at different timepoint compared to d0 were performed using paired Friedman test with a Dunn's multiple comparison post-test. The threshold for statistical significance was set to $p < 0.05$ (*$p < 0.05$; **$p < 0.01$; ***$p < 0.001$; ****$p < 0.0001$).

**Reporting summary**. Further information on research design is available in the Nature Research Reporting Summary linked to this article.

## Data availability

All data generated or analyzed during this study are included in this published article (and its supplementary information files).

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

## Acknowledgements

We thank the members of the COVIR team for their helpful suggestions. This work was funded by a grant from Paris-Saclay University ("Programme de recherche exceptionnel 2020 UPSaclay COVID-19", OSCOVID/OSVI). A.O. received a fellowship from the French Agency for HIV, Hepatitis and Emerging Diseases (ANRS-MIE). We thank B. Delache, F Relouzat, S. Langlois, Q. Sconosciuti, V. Magneron, P. Le Calvez, M. Potier, J. M. Robert, N; Dhooge T. Prot, and C. Dodan for the NHP experiments; L. Bossevot, M. Leonec, L. Moenne-Loccoz, M. Galpin-Lebreau, L. Pintore and J. Morin, M. Cavarelli, R. Marlin, M. Galhaut, and V. Contreras, Barendji, J. Dinh and E. Guyon for the NHP sample processing and discussions; S. Keyser for the transports organization; F. Ducancel, A. Pouget and Y. Gorin for their help with the logistics and safety management; B. Targat contributed to data management; We thank S. Van der Werf, S. Bellil and V. Enouf for contribution to viral stock challenge production and A. Nougairede for sharing the plasmid used for the sgRNA assays standardization. The Infectious Disease Models and Innovative Therapies (IDMIT) research infrastructure is supported by the "Programme Investissements d'Avenir", managed by the ANR under reference ANR-11-INBS-0008. The Fondation Bettencourt Schueller and the Region Ile-de-France contributed to the implementation of IDMIT's facilities and imaging technologies. The NHP model of SARS-CoV-2 infection have been developed thanks to the support from REACTing, the Fondation pour la Recherche Medicale (FRM; AM-CoV-Path) and the European Infrastructure TRANSVAC2 (730964).

## Author contributions

A.O., R.M., P.M., J.L., L.B., T.L., C.M., S.B. and C.B. performed the experiments: A.O. and C.B. characterized the immune profile of AT samples, analyzed and interpreted the data, and wrote the manuscrit. R.M., P.M., R.L.G., and N.D.B. conceived the experimental design for the SARS-CoV-2 infection in NHPs. G.P. and S.B. collected human AT samples during abdominal surgery. O.L., N.D.B., and C.B. designed the work on the adipose tissue samples. L.B. and N.D.B. have done viral load quantification and analysis on AT samples. R.M., L.B. and J.L. performed the Luminex analysis. T.L. performed the immunochemistry analysis. C.B. have drafted the work.

## Competing interests

The authors declare no competing interests.
