## [Peer Review File · Communications Biology]

Reviewers' comments:

Reviewer #1 (Remarks to the Author):

This is a timely study in area of SarsCov2 and adipose tissue
The data presented are clear and there is a good discussion and interpretation of the findings
Major comments

1. study model was mild COVID and thus has limitations in extrapolation of findings to humans.
2. the study duration was 7 days and thus cannot answer the larger question of longerterm sequestration of virus in AT and its contribution to cytokine storm or long covid
3. Clear evidence of virus in SCAT versus VAT/EAT compartments is presented and this should provide impetus for further human studies
4. is it possible that EAT data is explained by transient lung infection and thus reduced possibility for local spread of virus to epicardial fat
5. ACE2 data is interesting and hypothesis generating but somewhat confusing. If ACE2 is internalized on SarsCoV 2 contact would the ACE2 levels not be decreased in SCAT versus VAT. perhaps measurement of ACE2 at protein level would help here.
6. The lack of absolute T cell counts makes proportional assessment of T cell populations challenging. What does the reduced CD69 expression represent.. downregulation, shedding or a change in population? Are there soluble markers that could be measured to address this question. It would be interesting to assess whether this CD69 reduction has an affect on Tregs as well as cytokines such as IFN γ , IL17 and IL22, Gal 1 a CD69 ligand would also be of interest with respect to signaling.

Reviewer #2 (Remarks to the Author):

I have carefully read the article entitled " Detection of SARS-CoV-2 in subcutaneous fat but not visceral fat, and the disruption of fat-lymphocyte homeostasis in the macaque" by Olivo and colleagues. It is a descriptive article, demonstrating the presence of SARS-Cov2 within the adipose tissue of the infected host. A great strength of this study is that the authors were able to use the macaque model. This model allows the results to be transposed to humans with greater confidence.

Surprisingly, the virus does not target all adipose tissues but mainly the subcutaneous adipose tissue. The authors then show that its presence in this tissue induces immune disorders in lymphocytes, in particular phenotypic disorders which suggest functional consequences. Despite its descriptive nature, this work provides important information on an essential subject given the considerable impact that sars-cov2 has on overweight individuals. However, reading this article gives the feeling that the authors could have obtained much more data.

Major comments:

- 1 - I liked the introduction of the work, it is understood that the animal cohort has already been used in other published works. However, several elements are introduced, such as the impact that leptin can have on inflammation in relation to obesity. This cytokine is not studied in the study although it could provide important information.
- 2 - In a similar way, in the results section, the authors mention the importance of ACE2 and tmprss2 in the SARS viral cycle but only study ACE2, and omit to describe tmprss2. Why not detail the expression of tmprss2? The authors must already have the cDNA to perform qPCR...
- 3 - The authors show unambiguously the presence of the virus in SCAT, but what is the coronavirus positive fraction? Adipocytes, pre-adipocytes, mesenchymal stem cells, lymphocytes or macrophages?
- 4 - In the discussion section, the authors suggest a change in the behavior of SCAT lymphocytes in terms of migration, given the loss of CD69 expression. What about blood lymphocytes? Are they more numerous? What is their CD69 expression level?
- 5 - Finally, the question of the model could be discussed further in the discussion section. Indeed, the macaques used in this work are not overweight. Is there a possibility of having obese macaques? Could the proportion of adipose tissue in macaques influence the results obtained in this study?

Minor comments:

- 1 - Abstract: I think it would be wise to specify the role of CD69 in the abstract
- 2 - The last sentence of the abstract is unclear and should be reworded in my opinion.
- 3 - Figure 2: The use of a star to represent data related to macaque #5 is confusing. I thought it was the representation of the p-value.
- 4 - Figure 2a and 2b are redundant and should be merged.

Dear reviewers, dear members of the editorial board,

We first want to thank the reviewers and the editors for their interest in our work and for the opportunity to revise the manuscript. Although we were limited in the quantity of material available, we tried to add some additional data. We hope you will find this revised version of the manuscript suitable for publication in Communications Biology.

The editor identified three important points to clarify and improve the manuscript: (i) measure the ACE2 and TMPRSS2 at RNA and protein levels, (ii) provide absolute T cell counts, and (iii) establish which cells within SCAT are virus-positive.

Concerning the **expression of ACE-2 and TMPRSS2**, we apologize for the missing information. We indeed checked for TMPRSS-2 expression in adipose tissues (AT) in the macaque model, but we did not detect any mRNA coding for TMPRSS2 in all AT samples. We included this information in the manuscript. Concerning the regulation of ACE-2 expression, we agree that our formulation was misleading, as mentioned by one reviewer. We meant to stress the fact that the analyses at the mRNA level may not be representative of a change in the net expression of ACE-2 per adipocyte for different reasons: (i) because SARS-CoV binding to ACE-2 may modulate ACE-2 expression as described for SARS-CoV (Kuba et al., 2005) and SARS-CoV-2 more recently (Kuba et al., 2021; Yamaguchi et al., 2021) at the protein level; (ii) Moreover, as the density of adipocyte may differ in SCAT and VAT (Ibrahim, 2009), the difference of ACE2 mRNA in SCAT and VAT may also refer to a different proportion of adipocyte. VAT contains a larger number of inflammatory and immune cells (which may dilute the mRNA expression for gene coded in adipocytes within a larger proportion of nonmetabolic cells). So the main point was to point out the limited interpretation of the mRNA detection: higher expression of ACE-2 mRNA in SCAT may not reflect higher expression on a per cell basis, but rather a higher relative proportion of adipocyte among the complex cellular network that is AT and the difference in composition between SCAT versus VAT (El-Sayed Moustafa et al., 2020). Interestingly, a recent publication describes an increase in ACE-2 transcription levels in AT following COVID-19 infection (Basolo et al., 2022) whereas previous publication described a downregulation of ACE-2 following SARS-CoV binding in the lung at the protein level. This apparent contradiction may reflect a compensatory mechanism at the transcription level due to the downmodulation of ACE2 at the surface. One may thus hypothesize that the higher expression of ACE-2 in the SCAT is related to the persistence of SARS-CoV-2 in the SCAT in contrast to the VAT. However, and as previously mentioned, we did not observe clear difference between SARS-CoV-2 infected and noninfected animals. We have introduced a sentence in the text to better describe the potential limitation of the mRNA analyses performed on AT.

As suggested, we tried to determine ACE-2 expression at the protein levels. Unfortunately, no Ab currently available are cross-reactive with the macaca fascicularis model. In order to confirm the expression of ACE-2 in AT, we thus analyze human samples from non COVID patients. We have now inserted the immunohistochemistry study performed on SCAT and VAT from 5 patients not suffering from COVID-19 to confirm that ACE-2 is expressed at the protein level in AT (now in figure 2. D). We observed the expression of ACE-2 protein both in SCAT and VAT: 10 to 15% of adipocytes expressed ACE-2 (Figure 2.E). These proportions are probably underestimated as some adipocyte negative for ACE-2 on the reading field may express ACE-2 at a different depth. The ACE-2 expression was localized within the adipocyte cytoplasm, close to the nucleus. No difference in ACE-2 expression at the protein level was observed between SCAT and VAT of the SARS-CoV-2 uninfected subjects. We introduce these set of data in the manuscript and change the text accordingly.

The second point referred to the need to **provide absolute numbers** to better decipher the change occurring within the T cell compartment. We have now included the absolute numbers of cells counted in the stromal vascular fraction of AT per gram of tissue dissociated. At the level of the number, we collected similar number of stromal vascular cells in SARS-CoV-2 animals compared to the control groups. As the number of CD45⁺ cells is equivalent, the proportion of CD8 T cells reflects the numbers of CD8 T cells and

we will present only the value in proportion. However, to provide the maximum of information, we have now included the SVF number in the manuscript and have introduced a sentence in the text referring to the cell number.

The third point was referring to the **identification of the cell subset(s) within SCAT that are virus-positive**. Due to shortage of materials and the relatively limited number of infected cells, we were unable to provide clear answer about this question. Indirectly, the identification of ACE-2 protein preferentially on adipocytes clearly points towards adipocytes as crucial target of the infection. In the meantime, the literature in the field has now provided some important insight confirming the infection of adipocytes, both in vitro and ex vivo (Basolo et al., 2022; Martínez-Colón et al., 2021; Reiterer et al., 2021; Zickler et al., 2022), although the infection of AT macrophages has also been described in one report (Martínez-Colón et al., 2021). We inserted a sentence in the text to develop this important question of the cell types infected by SARS-CoV-2 in AT and introduced the references.

Finally, we also checked that our manuscript complies with the editorial policies as indicated in the guidance. Please find below a detailed response to the reviewers.

Responses to the reviewers

Reviewer #1:

This is a timely study in area of SarsCov2 and adipose tissue. The data presented are clear and there is a good discussion and interpretation of the findings

We thank the reviewer for his/her positive comment.

Major comments

1. Study model was mild COVID and thus has limitations in extrapolation of findings to humans.

We agree with the reviewer that this point should be emphasized. But we also believe that demonstrating that SCAT is also infected in a context of mild COVID is an important point, as it reveals that the presence of SARS-CoV-2 in the AT is not directly associated to severe form of COVID. We believe that our study thus is an important complement to study performed on samples collected during autopsy of patients having died from COVID.

2. The study duration was 7 days and thus cannot answer the larger question of long-term sequestration of virus in AT and its contribution to cytokine storm or long covid.

We agree with the reviewer that the question of the persistence of the virus is crucial. We had the opportunity to analyze the viral load in adipose tissue in animals sacrificed at day 43, and no viruses was detected in this batch of 6 male animals, suggesting the presence of the viral was transient in AT. Unfortunately, we did not collect sufficient quantity of adipose tissue to perform an immune characterization of the stromal-vascular fraction and evaluate whether the alteration of the immune compartment persisted. Because this batch of animals is male and not female, and because no immune characterization was performed, we just introduced a sentence in the manuscript describing this result, that suggest that there is no long-term sequestration of the virus in AT in a model of mild COVID.

3. Clear evidence of virus in SCAT versus VAT/EAT compartments is presented and this should provide impetus for further human studies

We thank the reviewer for this positive comment. We believe that this differential viral load is an important aspect. We tried to determine whether lung viral load may correlate with the AT viral load, but none was observed. Due to the small number of animals included in the study, we did not introduce this set of data in the current manuscript. However, we now provide a clearer representation of the data (each animal can be identified in all panels), so that this information is now indirectly available in the figure.

4. Is it possible that EAT data is explained by transient lung infection and thus reduced possibility for local spread of virus to epicardial fat

The striking difference in viral load between the different sites is probably the most surprising results. We discussed different hypothesis but we agree with that we should also introduce this notion of transient lung infection that reduce the possibility for local spread. We also introduce the notion that this restricted distribution may also differ in the context of severe infection. Alternately, one cannot exclude that VAT was infected but eliminated the virus.

5. ACE2 data is interesting and hypothesis generating but somewhat confusing. If ACE2 is internalized on SarsCoV-2 contact would the ACE2 levels not be decreased in SCAT versus VAT, perhaps measurement of ACE2 at protein level would help here.

We agree with the reviewer that the ACE2 data were confusing at the time and apologize for the misleading formulation. The neat difference in ACE-2 mRNA expression in SCAT than VAT and the heterogeneity in the level of expression were puzzling. As a point of discussion, we meant to stress the fact that the analyses at the mRNA level may not be representative of a change in the net expression of ACE-2 per adipocyte for different reasons: (i) because SARS-CoV binding to ACE-2 may modulate ACE-2 expression at the protein level as described for SARS-CoV (Kuba et al., 2005) and SARS-CoV-2 more recently (Kuba et al., 2021; Yamaguchi et al., 2021); (ii) Moreover, as the density of adipocyte may differ in SCAT and VAT, the difference within SCAT and VAT) may refer to a different proportion of adipocyte in both tissues. VAT is indeed described as a site that contains a higher proportion of hematopoietic cells (which may dilute the mRNA expression for gene coded in adipocytes among total cells). So the main point was to point out the limited interpretation of the mRNA detection: higher expression of ACE-2 mRNA in SCAT may not reflect higher expression on a per cell basis, but rather a higher relative proportion of adipocyte among the complex cellular network that is AT in SCAT versus VAT. A report has previously pointed out the impact of the cell composition of AT on ACE2 expression (El-Sayed Moustafa et al., 2020). This may also contribute to the high heterogeneity of ACE2 mRNA expression within SCAT.

Finally, a recent publication describes an increase in ACE-2 transcription levels in AT following COVID-19 infection (Basolo et al., 2022). One may thus conclude that the higher expression of ACE-2 in the SCAT is related to the persistence of SARS-CoV-2 in the SCAT in contrast to the VAT. As previously mentioned, we did not observe clear difference between SARS-CoV-2 infected and noninfected animals. We have introduced a sentence in the text to better describe the potential limitation of the mRNA analyses performed on AT and limit its interpretation to the detection of ACE-2 in the SCAT.

6. The lack of absolute T cell counts makes proportional assessment of T cell populations challenging. What does the reduced CD69 expression represent? Down-regulation, shedding or a change in population? Are there soluble markers that could be measured to address this question. It would be interesting to assess whether this CD69 reduction has an effect on Tregs as well as cytokines such as IFN γ , IL17 and IL22, Gal 1 a CD69 ligand would also be of interest with respect to signaling.

We have now included the absolute numbers of cells counted in the stromal vascular fraction of AT per gram of tissue dissociated. At the level of the number, we collected similar number of stromal vascular cells in SARS-CoV-2 animals compared to the control groups. As the number of CD45⁺ cells is equivalent, the proportion of CD8 T cells reflects the numbers of CD8 T cells and we will present only the value in proportion. To provide the maximum of information, we included the SVF number in the manuscript (Figure 3A) and introduced a sentence in the text referring to the cell number.

We thank the reviewer for the constructive comment about the expression and regulation of CD69 and its potential functional consequences. The reviewer provides a stimulating hypothesis to explore. We have indeed mostly addressed its potential impact on migration but did not explore the role of CD69 on the metabolic activity of CD69 bearing cells (Cibrián and Sánchez-Madrid, 2017; Cibrián et al., 2016), which could be especially relevant in the metabolic tissue that is AT. We agree that this aspect would be extremely important to explore in the future but will require combination of metabolic and immune

characterization that is not possible to perform for this current study. It is nevertheless an interesting prospect, notably in healthy conditions to evaluate the role of CD69 expression on the functional activity of AT T cells, with a potential regulation of the metabolic activity leading to a specific functional profile. We introduced a sentence about the potential impact of CD69 regulation on the metabolic activity of AT T cells, an aspect that was missing in the current manuscript.

We thank the reviewer for his/her interest in our work, and his/her constructive comment.

Reviewer #2:

Despite its descriptive nature, this work provides important information on an essential subject given the considerable impact that sars-cov2 has on overweight individuals. However, reading this article gives the feeling that the authors could have obtained much more data.

We thank the reviewer for his/her positive comment and acknowledge the descriptive nature of the work, and also the technical limitations of the study.

Major comments:

1 – The leptin is not studied in the study although it could provide important information.

We agree that we provided a limited set of information on the inflammatory and metabolic profile associated with the infection. We have now inserted some data about the inflammatory responses and the metabolic profile of the animals in the blood. The figure 1 has been expanded to introduce in C, some kinetics of cytokine levels before and following the infection. In this model of mild infection, we did not observe any significant change in IL-6 and TNF- α , cytokines that are produced by AT, nor IL-1 β . Interestingly, we observed a trend towards an increase in the concentration of MCP-1 at day 2 post-infection, but did not last as value at day 4 returned to baseline.

Regarding metabolic parameters (now introduced in Figure 1.D), we measured the concentration of leptin, adiponectin and insulin at baseline and at day 7 post-infection. We observed a trend towards a decrease in leptin associated with an increase in adiponectin and insulin. These data are in accordance with a change in the endocrine activity of AT, in favor of a skewing towards an anti-inflammatory profile. However, it may also reflect a transient change in food uptake, leading to decreased level of leptin. No change in food uptake has been noted but this question was not directly addressed. No drastic weight loss (2.5% weight loss) was noted. When confronting our results to the literature, we noted some important discrepancy. Regarding leptin, reports described higher level of leptin in the blood of SARS-CoV-2 infected patients (Larsson et al., 2021; van der Voort et al., 2020) compared to control groups. The patients included in these studies exhibited severe COVID in contrast to the disease developing in macaque. This discrepancy may support a role for leptin in the severity of the disease. Regarding adiponectin, very heterogeneous set of results have been reported showing either lower level of adiponectin (Kearns et al., 2021; Reiterer et al., 2021) or no difference (Blot et al., 2021; Van Zelst et al., 2020), or higher level of adiponectin (Caterino et al., 2021) in clinical contexts. An interesting work of Di Filippo et al favors the analysis of the adiponectin/leptin ratio (Di Filippo et al., 2021). They describe a low adiponectin/leptin ratio in mild COVID. Such mechanism may constitute a compensatory response to systemic inflammation that is blunted in more severe contexts.

Similarly, the trend towards an increase in insulin we observed in this model at 7 dpi does not fit with the opposite observation of the direct targeting of β pancreatic cells by SARS-CoV-2 leading to insulin decrease ex vivo (Tang et al., 2021; Wu et al., 2021). Although resistance to insulin has been described in patients with severe COVID (Armeni et al., 2020; Reiterer et al., 2021), limited data are available in mild COVID.

Finally, we tried to perform qPCR to assess the change in leptin and adiponectin mRNA at the level of the AT. As shown in figure R1, we observed both a trend towards a decrease in leptin and adiponectin in both SCAT and VAT. This observation is unexpected, as leptin and adiponectin are expected to be inversely regulated. However, and as previously described, this result may indicate that AT during infection is associated with a change in AT cell composition that thus reduce the mRNA level of these two adipokines produced by the adipocytes. To better characterize the metabolic equilibrium in AT, we analyzed the ratio of leptin/adiponectin production. No significant difference was observed but a trend toward an increase in the leptin/adiponectin ratio in the SCAT was observed in contrast to the increase in lep/Adi ratio in the VAT. As VAT is considered has the AT producing preferentially leptin, the trend towards a decrease in the blood may be related to the decrease in production in the VAT.

Because the data are not statistically relevant, presumably due to the low number of samples, we did not include this set of data in the manuscript, although the notion of a different metabolic change in SCAT and VAT during SARS-CoV-2 is an interesting insight that need to be further evaluated.

Figure R1 Adiponectin and leptin mRNA detection in AT.

To evaluate the metabolic changes occurring in AT during SARS-CoV-2 infection, we analyzed the expression of MRNA coding for adiponectin and leptin. PPIA was used as housekeeping gene. No significant change was observed indicating that no drastic change in the adiponectin/leptin balance is induced in this context of mild SRAS-CoV-2 infection.

Studies were performed on 5 uninfected female animals (including 3 SCAT and 5 VAT) and 5 SARS-CoV-2 infected female animals.

2 – Why not detail the expression of tmprss2?

We apologize for this missing piece of information. We have indeed tested by qPCR the expression of TMPRSS2, but no signal was detected in both subcutaneous and visceral AT. We introduced a sentence in the text confirming that no TMPRSS2 mRNA was detected in all the samples tested.

3 – The authors show unambiguously the presence of the virus in SCAT, but what is the coronavirus positive fraction? Adipocytes, pre-adipocytes, mesenchymal stem cells, lymphocytes or macrophages?

Due to shortage of materials and the relatively limited number of infected cells, we were unable to provide clear answer about this question. Indirectly, the identification of ACE-2 protein preferentially on adipocytes clearly points towards adipocytes as crucial target of the infection. In the meantime, the literature in the field has now provided some important insight confirming the infection of adipocytes, both in vitro and ex vivo (Basolo et al., 2022; Martínez-Colón et al., 2021; Reiterer et al., 2021; Zickler et al., 2022), although the infection of AT macrophages has also been described (Martínez-Colón et al., 2021). We inserted a sentence

in the text to develop this important question of the cell types infected by SARS-CoV-2 in AT and introduced the references.

4 – In the discussion section, the authors suggest a change in the behavior of SCAT lymphocytes in terms of migration, given the loss of CD69 expression. What about blood lymphocytes? Are they more numerous? What is their CD69 expression level?

The expression of CD69 is low in the blood, both in healthy conditions and in the SARS-CoV-2 context. Interestingly, in SARS-CoV-2 infected animals, we observed an increase of CD69+ CD4 T cells associated with a decrease in CD69+ CD8 T cells. There was no lymphopenia observed at d7 in the blood in the SARS-CoV-2 infected animals although lymphopenia appeared earlier (2 dpi), as commonly described. It is tempting to read the increase in the proportion of CD69+ cells among CD4 T cells as a trace of a potential mobilization of AT T cells to the blood. However, CD69 is primarily described as a marker of activation in the periphery. In favor of the hypothesis of a remobilization of CD69+ cells, we did not observe a concomitant increase in HLA-DR expression associated, suggesting that the increase in CD69+ is not related to ongoing activation. Additionally, CD8 T cells in the blood did not exhibit a similar increase in the proportion of CD69+ cells, which is also in favor with limited ongoing activation. The reason why such phenomenon is not observed for CD8 T cells may reflect a more direct migration towards site of viral persistence and/or different mobilization properties of AT CD4 and CD8 T cells. In contrast, the restored lymphocytes count may reflect a replenishment of blood compartment, but the contribution of AT T cells remains to be determined. We have now introduced the information about CD69 expression in the blood both as a supplemental figure and in the text, but did not extensively comment on these data, as they need further evaluation.

5 - Finally, the question of the model could be discussed further in the discussion section. Indeed, the macaques used in this work are not overweight. Is there a possibility of having obese macaques? Could the proportion of adipose tissue in macaques influence the results obtained in this study?

We agree that this is an important point. As mentioned, this work is performed on nonobese animals, and one may expect a more severe, or at least a different, set of alterations of the AT composition. We try to determine association between the weight of the animals and the major parameters of the infection (VL in the fluid, in AT), markers of inflammation (MCP-1) and immune changes in AT. The animals included in the study exhibited limited heterogeneity in terms of body weight and viral, inflammatory and immune parameters thus precluding any conclusion.

Minor comments:

1 - Abstract: I think it would be wise to specify the role of CD69 in the abstract

2 - The last sentence of the abstract is unclear and should be reworded in my opinion.

We have changed the abstract accordingly

3 - Figure 2: The use of a star to represent data related to macaque #5 is confusing. I thought it was the representation of the p-value.

We have changed the figures accordingly

4 - Figure 2a and 2b are redundant and should be merged.

Fig. 2a and 2b were separated because the detection was performed on two different sets of samples: detection of VL in fluids, versus detection in cell lysates. We believe it is important to keep the two graphs separated as the direct comparison of the VL would be misleading. We change the graph to make appear this difference in the technical settings.

We thank the reviewer for his/her interest in our work, and his/her constructive comments.

Protracted ketonaemia in hyperglycaemic emergencies in COVID-19: a retrospective case series. *Lancet. Diabetes Endocrinol.* **8**, 660–663.

Basolo, A., Poma, A.M., Bonuccelli, D., Proietti, A., Macerola, E., Ugolini, C., Torregrossa, L., Giannini, R., Vignali, P., Basolo, F., et al. (2022). Adipose tissue in COVID-19: detection of SARS-CoV-2 in adipocytes and activation of the interferon-alpha response. *J. Endocrinol. Invest.*

Blot, M., David Masson, Nguyen, M., Bourredjem, A., Andreu, P., Aptel, F., Barben, J., Beltramo, G., Bielefeld, P., Bonniaud, P., et al. (2021). Are adipokines the missing link between obesity, immune response, and outcomes in severe COVID-19? *Int. J. Obes. (Lond).* **45**, 2126–2131.

Caterino, M., Gelzo, M., Sol, S., Fedele, R., Annunziata, A., Calabrese, C., Fiorentino, G., D’Abbraccio, M., Dell’Isola, C., Fusco, F.M., et al. (2021). Dysregulation of lipid metabolism and pathological inflammation in patients with COVID-19. *Sci. Rep.* **11**.

Cibrian, D., Saiz, M.L., De La Fuente, H., Sánchez-Díaz, R., Moreno-Gonzalo, O., Jorge, I., Ferrarini, A., Vázquez, J., Punzón, C., Fresno, M., et al. (2016). CD69 controls the uptake of L-tryptophan through LAT1-CD98 and AhR-dependent secretion of IL-22 in psoriasis. *Nat. Im.* **17**, 985–996.

Cibrián, D., and Sánchez-Madrid, F. (2017). CD69: from activation marker to metabolic gatekeeper. *Eur. J. Immunol.* **47**, 946–953.

El-Sayed Moustafa, J.S., Jackson, A.U., Brotman, S.M., Guan, L., Villicaña, S., Roberts, A.L., Zito, A., Bonnycastle, L., Erdos, M.R., Narisu, N., et al. (2020). ACE2 expression in adipose tissue is associated with COVID-19 cardio-metabolic risk factors and cell type composition. *MedRxiv Prepr. Serv. Heal. Sci.*

Di Filippo, L., De Lorenzo, R., Sciorati, C., Capobianco, A., Lorè, N.I., Giustina, A., Manfredi, A.A., Rovere-Querini, P., and Conte, C. (2021). Adiponectin to leptin ratio reflects inflammatory burden and survival in COVID-19. *Diabetes Metab.* **47**.

Ibrahim, M.M. (2009). Subcutaneous and visceral adipose tissue: Structural and functional differences. *Obes. Rev.* **11**, 11–18.

Kearns, S.M., Ahern, K.W., Patrie, J.T., Horton, W.B., Harris, T.E., and Kadl, A. (2021). Reduced adiponectin levels in patients with COVID-19 acute respiratory failure: A case-control study. *Physiol. Rep.* **9**.

Kuba, K., Imai, Y., Rao, S., Gao, H., Guo, F., Guan, B., Huan, Y., Yang, P., Zhang, Y., Deng, W., et al. (2005). A crucial role of angiotensin converting enzyme 2 (ACE2) in SARS coronavirus-induced lung injury. *Nat. Med.* **11**, 875–879.

Kuba, K., Yamaguchi, T., and Penninger, J.M. (2021). Angiotensin-Converting Enzyme 2 (ACE2) in the Pathogenesis of ARDS in COVID-19. *Front. Immunol.* **12**.

Larsson, A., Lipcsey, M., Hultström, M., Frithiof, R., and Eriksson, M. (2021). Plasma Leptin Is Increased in Intensive Care Patients with COVID-19-An Investigation Performed in the PronMed-Cohort. *Biomedicines* **10**.

Martínez-Colón, G.J., Ratnasiri, K., Chen, H., Jiang, S., Zanley, E., Rustagi, A., Verma, R., Chen, H., Andrews, J.R., Mertz, K.D., et al. (2021). SARS-CoV-2 infects human adipose tissue and elicits an inflammatory response consistent with severe COVID-19. *BioRxiv* 2021.10.24.465626.

Reiterer, M., Rajan, M., Gómez-Banoy, N., Lau, J.D., Gomez-Escobar, L.G., Ma, L., Gilani, A., Alvarez-Mulett, S., Sholle, E.T., Chandar, V., et al. (2021). Hyperglycemia in acute COVID-19 is characterized by insulin resistance and adipose tissue infectivity by SARS-CoV-2. *Cell Metab.* **33**, 2484.

Tang, X., Uhl, S., Zhang, T., Xue, D., Li, B., Vandana, J.J., Acklin, J.A., Bonnycastle, L.L., Narisu, N., Erdos, M.R., et al. (2021). SARS-CoV-2 infection induces beta cell transdifferentiation. *Cell Metab.* **33**, 1577-1591.e7.

van der Voort, P.H.J., Moser, J., Zandstra, D.F., Muller Kobold, A.C., Knoester, M., Calkhoven, C.F., Hamming, I., and van Meurs, M. (2020). Leptin levels in SARS-CoV-2 infection related respiratory failure: A cross-sectional study and a pathophysiological framework on the role of fat tissue. *Heliyon* **6**.

Wu, C.T., Lidsky, P. V., Xiao, Y., Lee, I.T., Cheng, R., Nakayama, T., Jiang, S., Demeter, J., Bevacqua, R.J., Chang, C.A., et al. (2021). SARS-CoV-2 infects human pancreatic β cells and elicits β cell impairment. *Cell Metab.* **33**, 1565-1576.e5.

Yamaguchi, T., Hoshizaki, M., Minato, T., Nirasawa, S., Asaka, M.N., Niiyama, M., Imai, M., Uda, A., Chan, J.F.W., Takahashi, S., et al. (2021). ACE2-like carboxypeptidase B38-CAP protects from SARS-CoV-2-induced lung injury. *Nat. Commun.* **12**.

Van Zelst, C.M., Janssen, M.L., Pouw, N., Birnie, E., Castro Cabezas, M., and Braunstahl, G.J. (2020). Analyses of abdominal adiposity and metabolic syndrome as risk factors for respiratory distress in COVID-19. *BMJ Open Respir. Res.* **7**.

Zickler, M., Stanelle-Bertram, S., Ehret, S., Heinrich, F., Lange, P., Schaumburg, B., Kouassi, N.M., Beck, S., Jaeckstein, M.Y., Mann, O., et al. (2022). Replication of SARS-CoV-2 in adipose tissue determines organ and systemic lipid metabolism in hamsters and humans. *Cell Metab.* **34**, 1–2.

REVIEWERS' COMMENTS:

Reviewer #1 (Remarks to the Author):

The revised manuscript and response by authors has satisfactorily addressed this reviewers concerns

Reviewer #2 (Remarks to the Author):

The authors did a good job in this review process.